# Development of a Hydrophobic Carbon Sponge Nanocomposite for Oil Spill Cleanup

**DOI:** 10.3390/ma15238389

**Published:** 2022-11-25

**Authors:** Malika Medjahdi, Badra Mahida, Nouredine Benderdouche, Belaid Mechab, Benaouda Bestani, Laurence Reinert, Laurent Duclaux, Dominique Baillis

**Affiliations:** 1LGPME, Sidi Bel Abbes University, Sidi Bel Abbes 22000, Algeria; 2LSEA2M, Abdelhamid Ibn Badis University, Mostaganem 27000, Algeria; 3LMPM, Sidi Bel Abbes University, Sidi Bel Abbes 22000, Algeria; 4EDYTEM Laboratory, University of Savoie-Mont Blanc, 73000 Chambery, France; 5Univ Lyon, INSA Lyon, CNRS UMR5259, LaMCoS, 69621 Villeurbanne, France

**Keywords:** polyurethane, carbon-nanotubes, hydrophobic character, oil spill

## Abstract

Oil leaks (or spills) into the aquatic environment are considered a natural disaster and a severe environmental problem for the entire planet. Samples of polyurethane (PU) composites were prepared with high specific surface area carbon nanotubes (CNT) to investigate crude oil sorption. Scanning electron microscopy (SEM), Fourier-transform infrared spectroscopy (FT-IR), density measurements, and mechanical compression tests were used to characterize the polyurethane-carbon PU–CNT prepared samples. The spongy composites exhibited good mechanical behavior and a contact angle of up to 119°. The oleophilic character resulted in increased hydrophobicity, a homogeneous oil distribution inside the sponge, and a sorption capacity in a water/oil mixture of 41.82 g/g. Stress-strain curves of the prepared samples showed the good mechanical properties of the sponge, which maintained its stability after more than six sorption desorption cycles. The CNT–PU composites may prove very effective in solving oil pollution problems.

## Introduction

Oil leaks from shipwrecks have recently resulted in massive oil spills that have caused significant damage and put wildlife at great risk [1,2,3]. Combustion [1], in-situ chemical treatment [2], biological means [3,4], and physical adsorption are a few oil spill response methods that have been used [5]

Among these techniques, physical adsorption is widely used, owing to its high efficiency in cleaning up oil-contaminated waters [6] and low-cost. Since natural adsorbents [3,7] such as coals [8], clays [5], cotton fibers [9], zeolites [10], wool fibers [11], and sawdust [12] have low selectivity for hydrocarbons and low recovery and recycling rates, they have received much attention in research. However, their applications for hydrocarbon recovery are still quite limited.

In order to further increase the adsorption capacity of materials with high adsorption capacity and good selectivity, the introduction of these precursors as additives in the matrix of porous polymers [13], such as resins [14,15], foams [1,16], sponges [17,18], and aerogels [19,20,21] has also attracted interest. Polyurethane (PU) composites with honeycomb structure [22], PU–clay [23], PU–carbon [24], and PU–graphene [17,25] have recently been investigated for the separation of water–oil systems.

The good mechanical, thermal, and electrical properties of CNTs have increased their potential in a wide range of real applications [26,27]. For instance, they can be integrated into a polyurethane matrix to yield excellent hydrocarbon adsorbentsowing to their good hydrophobic character [28,29,30].

Recently, novel methods of mixing polyurethane and carbon nanotubes have been employed [31,32,33,34,35,36,37] to increase the number of pores and improve the hydrophobic character of the sponge in order to remove oils and other non-polar organic pollutants from water [38,39].

This work discusses the introduction of carbon nanotubes into the matrix of a polyurethane foam formulation without changing the properties of the polymer, based on FTIR analysis of the composite [40,41]. Wettability and contact angle measurements show the prepared material’s hydrophobic–oleophilic character. Furthermore, it has good flexibility, stability, and mechanical properties, with many pores according to optical microscopy analysis [42], which makes the modified foam a cost-effective multipurpose material that can effectively remove hydrocarbons, especially oil.

## 2. Materials and Methods

The polyurethane foam formulation procedure was detailed in previous work [24], mainly consisting of mixing toluene, diisocyanate, and polyols with some additives, including methylene chloride, stannous octoate, amines as catalysts, water as a blowing agent, and silicone as a surfactant (supplied by the “Style Mousse” mattress production company, Es senia, Oran, Algeria).

Table 1 shows the carbon nanotubes (CNTs) characteristics as supplied by Graphistrength Arkema, France.

The different steps for the preparation of the PU foam–CNT composite are as follows:The chemical components are incorporated in a given order (mixture A: 7.4 g of toluene diisocyanate (TDI) and 0.65 g of methylene chloride, mixture B: 10 g of polyol is thoroughly mixed with different weights of CNT for 15 to 20 s. 0.8 g of water, 0.2 g of silicone oil, 0.16 g of triethylenediamine, and 0.03 g of stannous octoate are mixed for 15 s at an agitation speed of 1000 rpm in a mixer (Figure 1).Ten seconds after the start of the operation, a chemical reaction occurs, releasing the inflating gas (CO_2_) produced from the reaction of TDI with water. The gas is diffused into the liquid and gives it a “creamy” consistency,At the same time as gas production, the foam begins to swell and increases viscosity. Finally, after about two minutes, maximum expansion occurs. At this stage, in a foam of suitable composition, the residual gas escapes through the top of the block, which has gained sufficient strength to maintain its shape.

The strength of the foam continues to rise, and after two and a half to three minutes, the foam ultimately sets.

At this time, the foam takes its final form and can be used after 24 h.

The whole procedure is performed under an ambient temperature (23 °C). The mold used is 4.2 cm in diameter and 21 cm in height.

The CNT was added to the PU formulation in the following w/w ratios in the preparation beaker, as shown in Table 2:

After thickening and cooling, the resulting sponge’s composites were unmolded and left in the open air for curing for 24 h. The prepared sponges were stored away for characterization and evaluation.

The prepared samples were characterizedfor:Their surface using an optical microscope apparatus: Hirox digital microscope KH 8700 at 100× magnification.The density and porosity were measured using a density determination kit for analytical balance. The following Equations (1) and(2) wereused to calculate the porosity:
(1)$$Porosity\left(\%\right)=\left(\frac{V_{pores}}{V_{apparent}}\right)\times 100=\left(\frac{V_{apparent}-V_{skeleton}}{V_{apparent}}\right)\times 100$$
(2)$$\mathrm{With}\;V_{skeleton}\;=\;\frac{m_{0}}{d_{skeleton}}$$
where:

m_0_ is the weight of the composite (g),

d_skeleton_ is the density of the composite sponges measured with the density determination kit (g/cm^3^),

V_skeleton_ is the volume occupied by the composite sponge skeleton/ pores (cm^3^).

V_pores_ is the volume representingall the pores of the composite sponges (cm^3^), and V_apparent_ is the volume obtained from the composite sponge dimensions (cm^3^).

FTIR spectroscopy was carried out using a NicoletFTIR spectrometer (400–4000 cm^−1^ range), Raman spectroscopy (Avantes, at 473 nm);The mechanical properties of the samples were performed according to the ISO 24999 standard using a Lloyd Instrument LF Plus 2745 (Ametek company) equipped with two parallel surfaces and a 100 N detection cell with 80% of strain at a controlled deformation speed of 5 mm/min;For wettability measurements, the characterization of wetting of prepared surfaces was carried out by an analysis of the process of intrusion of the water drops into the pores.

Software (ImageJ V 1.8.0 Manipulation Program) allowed the measurement of the baseline of each drop. Equation (3) was used to evaluate the contact angle:θ_w_ = 2 arctg (2 h/d)(3)
where:

θ_w_ is the Wenzel’s contact angle of the surface,

d and h are the two geometrical parameters of each fluid drop discussed above.

The measurements are repeatedeight to ten times to calculate the results.

A hydrostatic Mohr balance, with measurement accuracy to within 10^−4^ g, was used for oil sorption measurements.

The composite samples were immersed into a water-oil system for 90 min to ensure saturation and then picked out and weighed (see Figure 2).

The weight measurements were performed quickly and repeated in triplicate. The method applied for measuring the sorption capacity of hydrocarbons $Q_{\left(\frac{g}{g}\right)}$, in particular petroleum, is based on standard ASTMF726 99: standard method for testing the sorption capacity of adsorbents (Equation (4)).
(4)$$Q_{\left(\frac{g}{g}\right)}=\frac{m_{sorb}-m_{0}}{m_{0}}$$
where *m_sorb_* is the composite’s weight after sorption (g), and m_0_ is its initial weight (g).

The regeneration of these sorbents is based on a chemical method where the sorbents are immersed in toluene and then washed with petroleum ether three times. Then the samples are dried in an oven.

## 3. Results and Discussion

### 3.1. Morphology Analysis

Figure 3 shows the images obtained for the raw PU sponge and 0.212% CNT–PU sponge using an optical microscope apparatus. Regular and almost pentagonal alveolar closed structures are revealed. CNTs agglomerate at the inter-alveolar borders, whereas for higher CNT loading (Figure 4), the nanotubesagglomerated at a few points in the polymer matrix and were not uniformly distributed, as is necessary to promote effective reinforcement.

The use of higher amounts of filler resulted in a loss of the polyhedral morphology characteristic (Figure 3, image PU (d)) and the deterioration of the morphology of the sponge, as shown in Figure 4.

This result may be attributed to the poor dispersion of nanoparticles for their higher content.

### 3.2. Spectroscopic Analysis

FTIR spectra of the obtained polyurethane composites using a NicoletFTIR spectrometer (France) are shown in Figure 5.

The characteristic peaks of the PU are evident near 3293 cm^−1^ [43]. These peaks belong to the corresponding vibration of the hydroxyl functional group (O-H), probably due to their existence in an unreacted polyol (reaction of the polyol with isocyanate). In addition, the absorption of isocyanate does not appear at 2275 cm^−1^ due to the complete reaction of the isocyanate with the polyol and water to form urethane and carbon dioxide bonds, respectively [40]. The peaks near the wave number of 2950 cm^−1^ are associated with the vibration of the functional group in the carbon chains –CH_2_-and –CH [39,42]. The sharp peak at the wave number 1717 cm^−1^ is linked to the urethane function and the carbonyl (C = O) of the functional group existing in the urethane bond [41,44]. The peak at 1093 cm^−1^ is attributed to the etheric vibration of the C-O group [40,42]. The peak at 1531 cm^−1^ is due to the C-N and N-H groups (Amides II) [39]. The average intensity signal observed around 1596 cm−^1^ belongs to the aromatic vibration C = C [44,45]. A large alcoholic O-H peak is detected at 3293 cm^−1^ [43]. C-H of the aromatic cycle can be seen at the wavelength of 814 cm^−1^ [44,45].

The figure presented above shows the similarity of the spectra for PU and the three composites of PU-% CNT, probably due to the apolarity of the CNT.

As an example, the Raman and IR spectra of PU(b) were compared (see Figure 6) to obtain more information about new functions and secondary reactions. 

Both spectra presented in Figure 6 show that the loading of the PU–CNT treatment does not lead to any secondary reaction; only the characteristic peaks of the matrix and the charges do appear.

### 3.3. Density and Porosity Analysis

Density is an essential parameter of sponge performance regarding the durability of a flexible sponge (possibility of reuse). Therefore, to assess the rigidity of raw PU sponge and composites, it is essential to measure their density.

Figure 7 shows the density and porosity values obtained for the samples prepared.

It can be noted that porosity and density increase with carbon nanotube content. For a higher CNT content, the porosity increases significantly from about 90% to 94% and 97% for 0.106, 0.212% and 0.319 wt.% of CNTs, respectively.

CNTs do not react with the reagents during the preparation reaction; therefore, they are considered an additive filler that increases the mass of the sponges for the same volume and possibly causes a decrease in the volume for higher CNT loadings.

Javni et al. (2011) observed that nanoparticles increase the number of bubble sites with a reduced cell size compared to a neat polyurethane sponge, and consequently, the porosity increases [46,47]. The increase in porosity is due to the destruction of the PU skeleton by CNTs.

### 3.4. Mechanical Properties

Compression was carried out along the thickness axis with a controlled deformation speed.

The stress-strain curves, bending strength, and modulus values obtained are illustrated in Figure 8 and Figure 9.

An increase in the compressive strength is observed for composite samples with a small addition of CNTs (less than 0.3%) while higher CNT content causes a significant decrease in the mechanical strength. The same tendency was observed for Young’s modulus values.

The samples’ highest mechanical properties were measured with 0.212% addition of CNT where the compressive strength was 35 MPa and Young’s modulus 1282 MPa.

Such changes in the mechanical properties of the composite samples can be explained in terms of characteristic features of their structure as for a higher amount of CNTs, the time of mixing viscous polyol mixture with isocyanate is too short of obtaining a uniform material, which may also affect these properties [47].

#### 3.4.1. Wettability Behaviour

The contact angles of the prepared samples were measured to assess the oleophilic and hydrophobic character or wettability of polyurethane (PU) for oil spill cleanup.

Surface wettability measurements were performed using the sessile drop method. In these experiments, the samples were introduced into a glass chamber, and the liquid drop was deposited on the samples using a microsyringe (Figure 10).

Table 3 gives the left and right contact angles for different samples.

A contact angle (θ) of 101° was observed for the composites PU (a) while for PU (b) and PU (c), the angles were 119 and 129, respectively, whereas the neat PU sponge exhibited a more hydrophilic behavior with θ = 81°. The wettability of PU–CNT composites is partial for water and total for petroleum, which explains the oleophilic behavior of composites. This strong hydrophobic and oleophilic qualityof the samples prepared can be attributed to the surface chemistry having a high content of sp2 carbon (hydrophobic component) [21,48,49,50].

#### 3.4.2. Sorption Capacity

Due to crude oil (petroleum) volatility, sorption experiments were carried out for a 90 min contact time. Figure 11 shows the sorption isotherm of oil onto the CNT–PU samples investigated. It can be noted that saturation was reached within the first few minutes, i.e., around a quarter of an hour, for all PU–CNT composites, unlike raw PU. Oil sorption capacity onto the virgin PU sample attained 10.10 g/g, while for the samples with 0.319% CNT, it increased to 41.82 g/g due to higher porosity at the expense of mechanical strength. Samples with 0.106 and 0.212 CNT content performed better than the raw sponge, with 22.71 and 30.67 g/g for the first minutes of the sorption experiments, respectively.

The reusability of the prepared PU sponges after pressing them to exude oil was also tested in this work [51]. The sorption–desorption cycle investigation shows that approximately only 1.5% could not be desorbed (Figure 12), which pleads for the possible use of the prepared samples for petroleum spill cleanup.

## 4. Conclusions

CNT-filled polyurethane sponges were prepared to improve oil sorption for spill cleanup applications. In addition, the prepared (PU–CNT) compositesexhibited high porosity.

The results showedan increase in hydrophobic behavior with a contact angle of 119° ± 3° for the 0.212 wt.% CNT sample with a homogeneous distribution within the foam and a good oleophilic character with a crude oil sorption capacity in a water–oil mixture of 41.82 g/g. Reuse tests show that the PU composite remains mechanically stable for more than six sorption-desorption cycles.

The mechanical behavior of the synthesized composites showed that the elastic behavior of the (PU–CNT) composites increased with CNT content, with a Young’s modulus increasing from 444.44 MPa for the raw PU to 1282.05 MPa for PU with 0.212 wt.% CNT. However, a higher CNT content resulted in poor mechanical stability.

In conclusion, the PU–CNT composite manufacturing process presents a cost-effective way to solve oil pollution problems. Furthermore, the regeneration of the sample used by simple mechanical compression for oil recovery promotes the application of such composites in oil cleanup.

## Figures and Tables

**Figure 1 materials-15-08389-f001:**
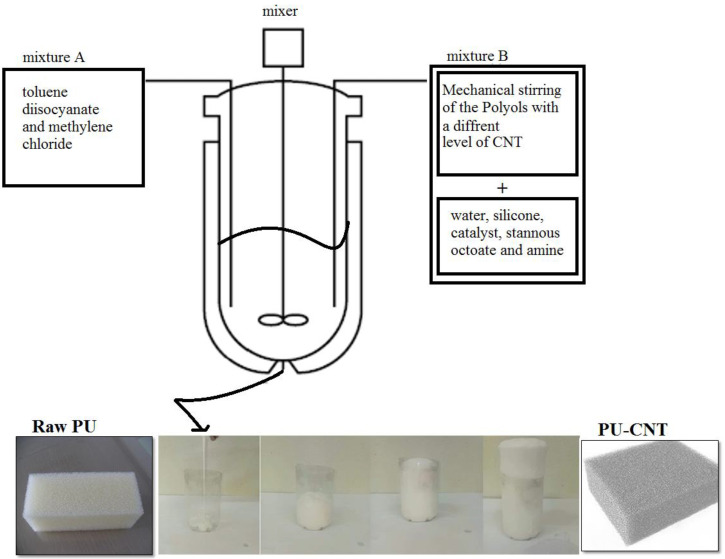
Preparation of raw PU foam and PU–CNT composite.

**Figure 2 materials-15-08389-f002:**
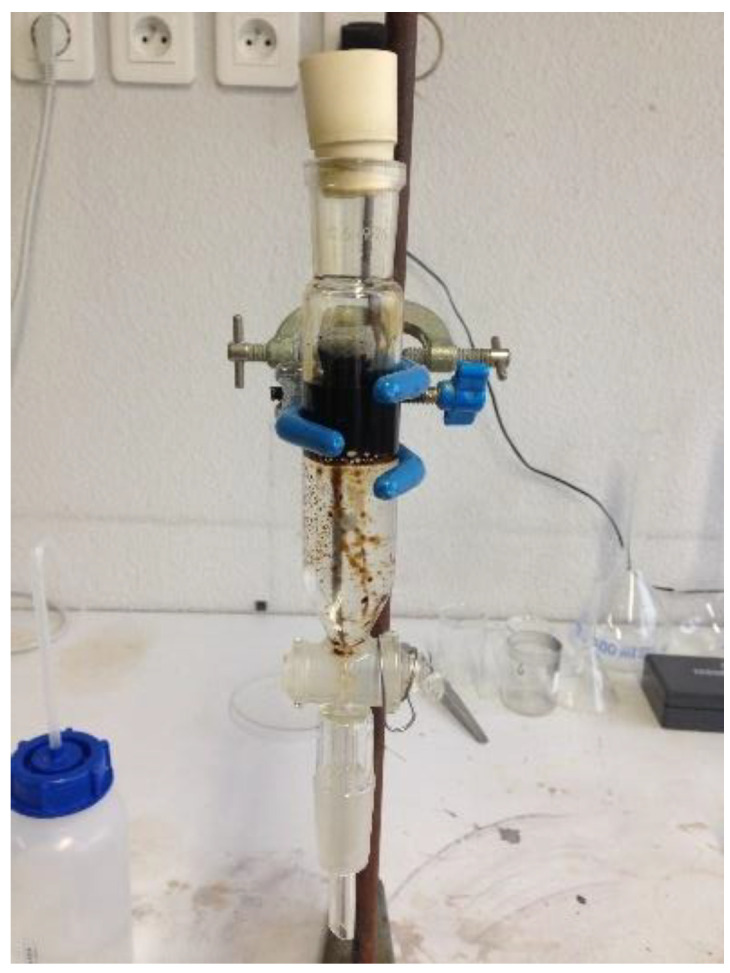
Oil sorption capacities test.

**Figure 3 materials-15-08389-f003:**
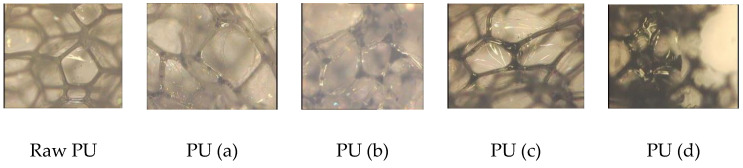
Optical microscope images of raw sponge and modified sponges at 100× magnification.

**Figure 4 materials-15-08389-f004:**
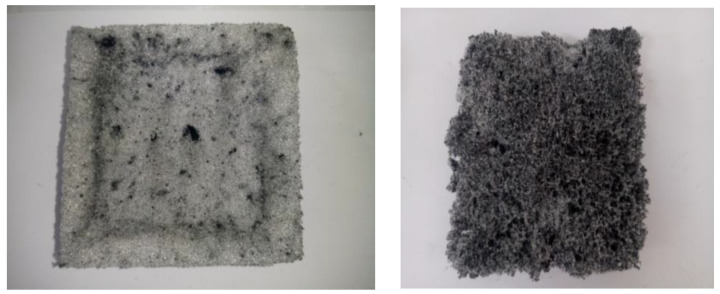
Higher % CNT–PU samples’ aspects.

**Figure 5 materials-15-08389-f005:**
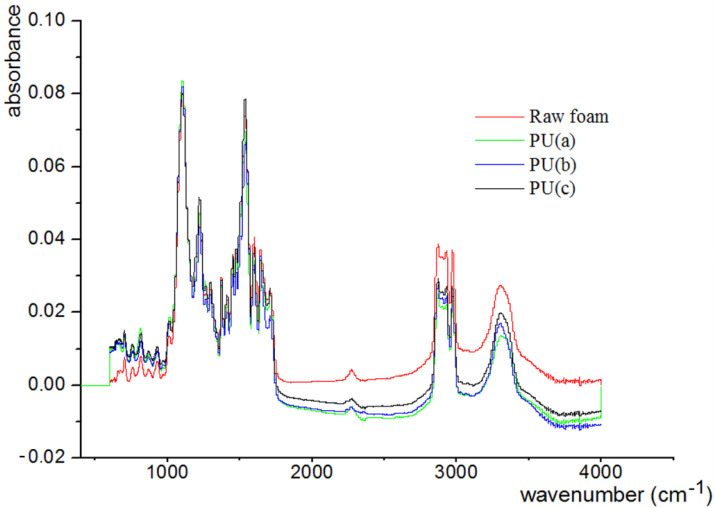
FTIR spectrum of raw and modified polyurethane.

**Figure 6 materials-15-08389-f006:**
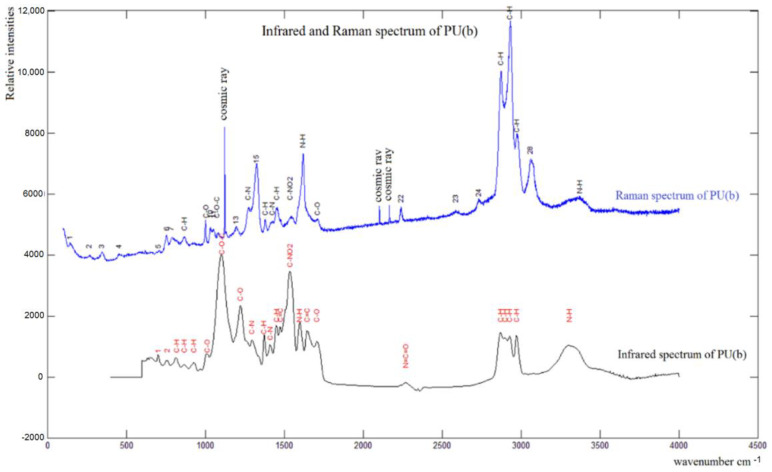
Raman andFTIR spectra of sample PU (b).

**Figure 7 materials-15-08389-f007:**
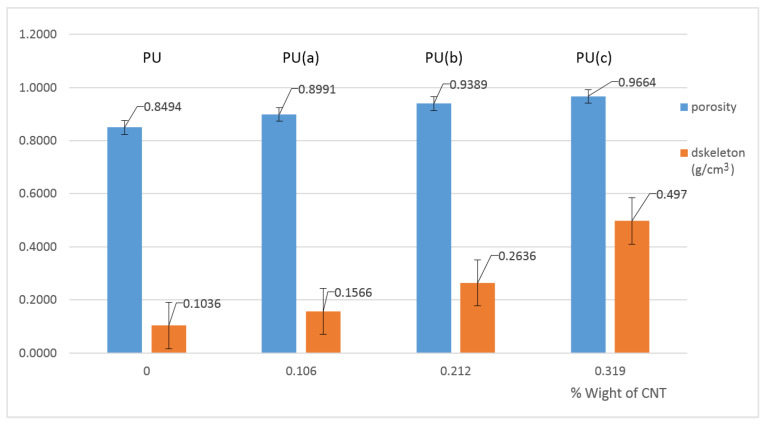
Density and porosity values for the CNT–PU composites prepared.

**Figure 8 materials-15-08389-f008:**
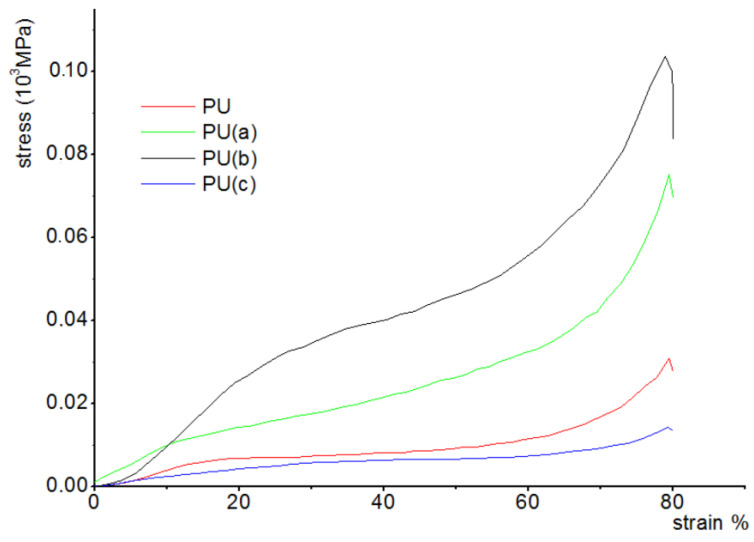
Stress–strain curves for the CNT–PU composites studied with 80% strain at a 5 mm/min speed.

**Figure 9 materials-15-08389-f009:**
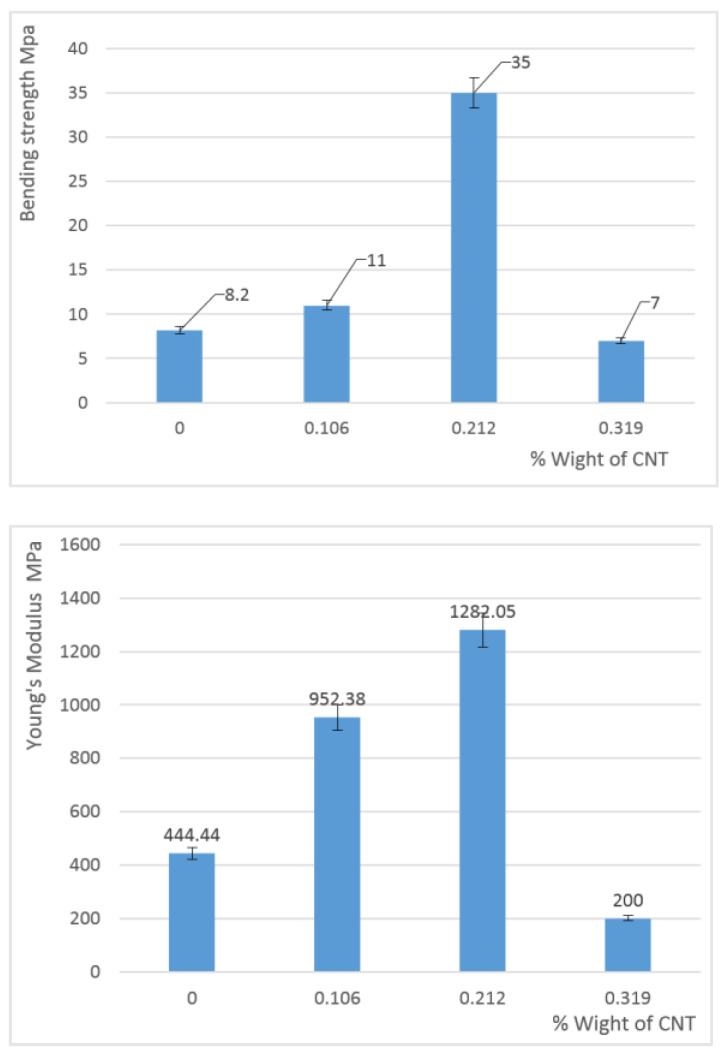
Bending strength and Young’s modulus values for the CNT–PU composites studied.

**Figure 10 materials-15-08389-f010:**
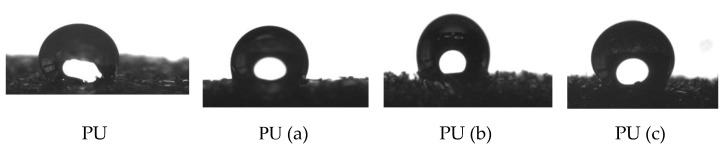
A drop of water on the surface of raw and modified PU.

**Figure 11 materials-15-08389-f011:**
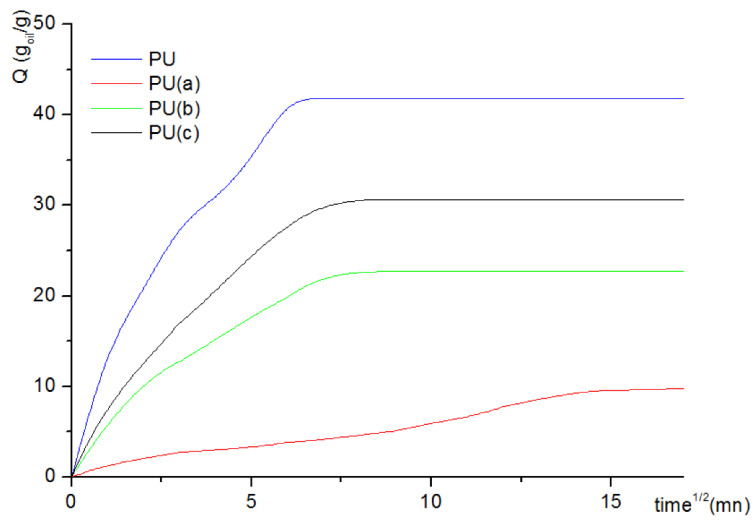
Oil uptake of raw PU and PU–CNT composites.

**Figure 12 materials-15-08389-f012:**
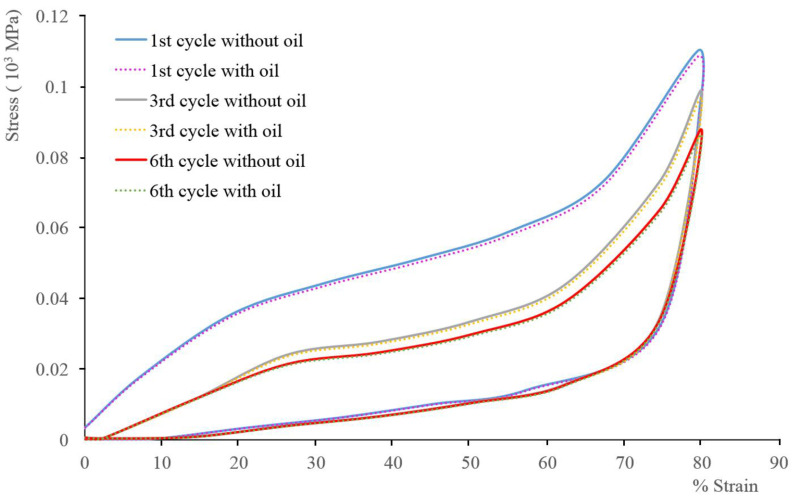
Mechanical behavior ofthe PU (b) sample with 80% strain at 5 mm/min speed, dry (full lines) or with oil.

**Table 1 materials-15-08389-t001:** Characteristics of carbon nanotubes.

Diameter of CNT	1.6 ± 0.4 nm
Length of CNT	≥5 µm
Density (g/cm^3^)	1.34
adsorption energies	Lower compared to graphite
Adsorbate-nanotube bonds	25 à 75% stronger compared to graphite
Conductivity	3.10^4^ S/cm
Resilience	100 times higher than steel
tensile strength	30 et 150 GPa
Young’s modulus	600–1200 GPa
Carbon nanotubes (CNT) content	≥80 wt.% CNT
	≤15 wt.% Metal impurities
	<5 wt.% Moisture

**Table 2 materials-15-08389-t002:** The weight percent of CNT added to PU formulation.

Weight Percent (%) CNT Added	Designation of Sponge
0.106	PU (a)
0.212	PU (b)
0.319	PU (c)
0.425	PU (d)
0.541	PU (e)

**Table 3 materials-15-08389-t003:** Left and right contact angles for raw and modified PU.

Samples	θ Left	θ Right	θ Contact Angle	Standard Deviation
PU	80.52	81.75	81.13	0.381
PU (a)	99.14	103.751	101.445	2.361
PU (b)	122.015	116.17	119.092	3.184
PU (c)	129.363	128.4	128.881	0.100

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
