# Peer review of "Development of a Hydrophobic Carbon Sponge Nanocomposite for Oil Spill Cleanup"

_materials, 2022, doi:10.3390/ma15238389_

Round 1

Reviewer 1 Report

The authors aim to story the CNT-filled-polyurethane adsorbent and its application in oil spill recovery. Unfortunately, I do not think this paper can be accepted for publication. Here are my comments and suggestions.

1, this manuscript has NOT been carefully checked before the author submitted it since there found many irregularities in expression. Besides, Figures 2, 5, 6, 7, 8, 9, and 10 look very coarse. And the caption of the X-axis in Figure 10 is doubted to be wrong. As to this point, the authors strongly suggest paying more attention before the next submission, whichever journal it will go to.

2, the author did not provide detailed information about the instrumentations, especially the conditions, which is not beneficial for experimental replication for the scientific community. For instance, in the discussion, we could see some physicochemical instrumentation was conducted in NOT-identical conditions, for example, FTIR.

3, What is the relationship between the images in Figure 4?

4. As searched online, one paper had been published before, also relative to polyurethane adsorbent yet modified by multi-walled CNT for the same purpose. Please refer to it and make a comparison. Besides, citing it as one advance of this specific field is necessary.

Author Response

Dear Reviewer,

We greatly appreciate the pertinent comments of Reviewer. We have modified the manuscript accordingly to try to improve the revised version.

R1/ All figures were revised and improved according to Reviewer suggestions.

2, the author did not provide detailed information about the instrumentations, especially the conditions, which is not beneficial for experimental replication for the scientific community. For instance, in the discussion, we could see some physicochemical instrumentation was conducted in NOT-identical conditions, for example, FTIR.

R2/ Information was added so as to allow replication.

3, What is the relationship between the images in Figure 4?

R3/ The samples presented in the images are loaded by a high rate of CNT which leads to a bad structural morphology

  1. As searched online, one paper had been published before, also relative to polyurethane adsorbent yet modified by multi-walled CNT for the same purpose. Please refer to it and make a comparison. Besides, citing it as one advance of this specific field is necessary.

R4/ There are many formulations of polyurethane described in the literature. In the paper, Reviewer has cited authors that chose the IDM with polyol as the polyurethane polymerization base reactant but the quantities used are generally larger than those using TDI with polyol for the same density also for the CNT ratios. In our study, we tried a formulation to prepare a cost-effective sorbent composite.

Reviewer 2 Report

This work has very good results to show, but they are not presented in a proper way to be sound. Some changes must be performed before the manuscript can be accepted for publication. Finally, authors should take care of some oversights. Generally it is a good work, but the appearence needs an improvement. Please see the attached pdf for my comments.

Author Response

Dear reviewer,

We would like to thank Reviewer for the very pertinent commenst and remarks. .

The changes made to the paper according to Reviewer’s comments in attached PDF are highlighted in red in manuscript.

In page 1,

  • (PU) is added
  • The space isadded
  • The bracket are removed
  • " as"is added
  • the paragraph " Carbon nanotubes (CNT) with their good hydrophobic character [19, 21] can be inte- grated into a polyurethane matrices to yield excellent hydrocarbon adsorbents [26 - 28]. " was replaced by " Carbon nanotubes (CNT) can be integrated into a polyurethane matrix to yield excellent hydrocarbon adsorbents [28 - 30], due to their good hydrophobic character."

In page 2,

  • space is added
  • the chemicals were supplied by the “Style Mousse” mattress production company , Es senia, Oran, Algeria
  • the ";" was replace by ","
  • "style mousse" is the name of mattress production company which supplied all the chemicals
  • the description of procedure was added:
  •  

REPRENDRE SELON LE TEXTE CORRIGE

  • The different steps for the preparation of the PU foam- CNT composite are as follows:

The chemical components are incorporated in a given order (mixture A: 7.4g of Toluene Diisocyanate (TDI) and 0.65 g methylene chloride, mixture B: 10 g of Polyol was fully mixed with deferent weight of CNT for 15 to 20 s. 0.8 g of water, 0.2 g of silicone oil, 0.16 g of triethylenediamine and 0.03 g stannous octoate are added and mixed for 15 s) and mixed for about five seconds at an agitation speed of 1000 rpm in a mixer (Figure 1).

Some ten seconds after the start of the operation, a chemical reaction occurs, releasing the inflating gas (CO2) produced from the reaction of TDI with water. The gas is diffused into the liquid and gives it a "creamy" consistency,

At the same time of gas production, the foam begins to swell and increases in viscosity. After about two minutes, maximum expansion occurs. At this stage, in a foam of suitable composition, the residual gas escapes through the top of the block that has gained sufficient strength to maintain its shape.

The strength of the foam continues to rise and after two and a half to three minutes, the foam completely sets.

At this time, the foam takes its final form and can be used after 24 hours.

The whole procedure was performed under ambient temperature (23 °C). The mold used was 4.2 cm in diameter and 21 cm in height.

In page 3,

  • the details were added to the paper:

The prepared samples were characterized for surface using an optical microscope apparatus, density using a Density Determination Kit for analytical balance, porosity, FTIR spectroscopy using a Nicolet FTIR spectrometer (400-4000 cm-1 range), Raman spectroscopy (473 nm), mechanical properties using a Lloyd Instrument LF Plus 2745 (Ametek company) equipped with two parallel at surfaces and a 100 N detection cell. Compression was carried out along the thickness axis with a controlled deformation speed of 5 mm/min and wettability using an analysis of the intrusion process of the water drops into the pores.

In page 4

  • Optical microscope Images of raw sponge and modified sponges at 100× magnification
  • images of PU(a), (b), (c) waere added
  • images were added

In page 5

  • space was removed
  • the labels were enlarged

In page 6

  • the labels were enlarged
  • correction was maded for subscript

in page 7

  • labels were enlarge

In page 8

  • "wight" was replace by "weight" in figure 09

In page 9

  • labels were enlarge
  • the sorption-desorption cycles without or with oil were added in revised paper

Reviewer 3 Report

The authors synthesized flexible Polyurethane (PU)/ carbon nanotube (CNT) sponge for oil spill clean-up application. The developed nanocomposite sponges were highly porous with good contact angle of up to 119°. More importantly, they exhibited a crude oil sorption capacity in water/ oil mixture of 41.82 g/g. Additionally, reusability tests suggested that the PU/CNT composite sponges remained mechanically stable for more than six sorption-desorption cycles. The work is interesting and can be published in Materials if the following issues can be addressed:

1. The abstracts need to be improved with more details about the performance of the composite sponges.

2. Abbreviations need to be clearly defined when they are first mentioned in the manuscript (ex. PU, CNT, SEM…)

3. The works of https://doi.org/10.1016/B978-0-12-812667-7.00001-X and https://doi.org/10.1016/B978-0-08-102722-6.00006-7 should be cited in the introduction for better review of carbon nanotube composite fabrication.  

4. In section 2, more information about the chemicals, equipment, and standard/methods used in this work needs to be provided.

5. Table 1 needs to be carefully checked. It has many incorrect information and spelling errors. For example, electrical conductivity of 3.104 S/cm is too low for normal CNTs.

6. All figures need to be improved with better legends, captions, scale bars, and spell check.

7. In section 2, more description is required for the fabrication steps of PU/CNT sponges.

8. Table 2 shows 5 types of PU/CNT composite. Why did all figures in this manuscript only present data of 2 or 3 types?

9. What are the differences between the 4 bottles in Figure 2? Why did a part of the 4th sample absorb oil?

10. The sentence of “CNTs do not react with the reagents during the preparation reaction, therefore they are considered as an additive filler that increases the mass of the sponges for the same volume and possibly causes a decrease in the volume for higher CNT loadings.” is not clear. If CNTs only increase mass of the sponges without volume change, how could it cause a decrease in the volume for higher CNT loading?

11. Error bars are required for Figures 7 and 9. Images in Figure 10 should have the same magnification for better comparison.  

12. Several grammar and spelling errors in the manuscript need to be corrected. The writing needs to be improved significantly.

Author Response

Dear reviewer,

We thank Reviewer for the valuable comments which helped us improve the manuscript.

In manuscript, the changes made to the paper according to Reviewer’s comments are highlighted in green.

Please find the answers below. We hope they are satisfactory.

Q1/ The abstracts need to be improved with more details about the performance of the composite sponges.

R1/ The abstract was improved with more details.

Q2/ Abbreviations need to be clearly defined when they are first mentioned in the manuscript (ex. PU, CNT, SEM…)

R2/ the abbreviations were clearly defined.

Q3/ The works of https://doi.org/10.1016/B978-0-12-812667-7.00001-X and https://doi.org/10.1016/B978-0-08-102722-6.00006-7 should be cited in the introduction for better review of carbon nanotube composite fabrication.  

R3/ The works of https://doi.org/10.1016/B978-0-12-812667-7.00001-X and https://doi.org/10.1016/B978-0-08-102722-6.00006-7 were cited in the introduction.

Q4/ In section 2, more information about the chemicals, equipment, and standard/methods used in this work needs to be provided.

R4/ More information was added.

Q5/ Table 1 needs to be carefully checked. It has many incorrect information and spelling errors. For example, electrical conductivity of 3.104 S/cm is too low for normal CNTs.

R5/ there was indeed a spelling error. Corrected .

Q6/ All figures need to be improved with better legends, captions, scale bars, and spell check.

R6/ All figures were improved with better legends, caption, scale bars and spell check.

Q7/ In section 2, more description is required for the fabrication steps of PU/CNT sponges.

R7/ Description of fabrication steps of PU/CNT sponges was added

  1. Table 2 shows 5 types of PU/CNT composite. Why did all figures in this manuscript only present data of 2 or 3 types?

R8/ Only three types were presented because of the poor quality due to the destruction of their structure.

Q9/ What are the differences between the 4 bottles in Figure 2? Why did a part of the 4th sample absorb oil?

R9/ Actually, we used for this study a closed system so Figure 2 was changed.

  1. The sentence of “CNTs do not react with the reagents during the preparation reaction, therefore they are considered as an additive filler that increases the mass of the sponges for the same volume and possibly causes a decrease in the volume for higher CNT loadings.” is not clear. If CNTs only increase mass of the sponges without volume change, how could it cause a decrease in the volume for higher CNT loading?

R10/ For the polyurethane polymerization reaction, the polyol reacts with one part of TDI and the other part reacts with water to produce CO2 the main agent for pore formation. The low % CNT is usually scratched on the corners of the pores and sometimes on the walls that do not change the volume because the foams can occupy the space we have (mold volume). For high % CNT can damage the pore walls and therefore the volume will be decreased

Q11/ Error bars are required for Figures 7 and 9. Images in Figure 10 should have the same magnification for better comparison.  

R11/ Error bars were added for Figures 7 and 9 and the Images in Figure 10 was magnified at same scale for comparison.

Q12/ Several grammar and spelling errors in the manuscript need to be corrected. The writing needs to be improved significantly.

R12/ Grammar and spelling errors were checked.

Round 2

Reviewer 1 Report

Regarding the technical demonstration, the authors need to check this manuscript word by word carefully and sentence by sentence. In addition, the references are undoubtedly not prepared well.

Author Response

Dear Reviewer,

We greatly appreciate the pertinent comments of Reviewer. We have modified the manuscript accordingly to try to improve the revised version.

The demonstration technique was reviewed meticulously according to Reviewer’s valuable suggestions. We hope the changes made are satisfactory. 

The references were verified and prepared using the software “Mendeley”

Some specific changes are cited below.

In the revised manuscript, the changes were highlighted by blue color.

Sincerely,

In page 1

currently” was deleted

“regarded as “was replaced by “considered”

“a” was added

“a lot of” was replaced by “much”

In page 2

“a large number of” was replaced by “many”

“The” was added

“a” was deleted

“a” was added

“fully” was replaced by “thoroughly”

“in” was delated

In page 11

Reviewer 2 Report

I should thank authors for taking in account most of my comments. I am happy to suggest the acceptance of the manuscript.

Author Response

Dear Reviewer,

We greatly appreciate your comment.

Sincerely